# Scientists as Story-tellers: the explanatory power of stories told about environmental crises Jenni Barclay[1], Richie E.A. Robertson[2], M. Teresa Armijos[3],

[1]School of Environmental Sciences, University of East Anglia, Norwich, NR4 7TJ, United Kingdom
Seismic Research Centre, University of the West Indies, St. Augustine, Trinidad
[3]School of International Development, University of East Anglia, NR4 7TJ, United Kingdom

*Correspondence to*: Jenni Barclay (J.Barclay@uea.ac.uk)

**Abstract.** This paper examines how storytelling functions to share and to shape knowledge, particularly when scientific knowledge is uncertain **because of rapid environmental change.** Narratives or stories are the descriptive sequencing of events to make a point. In comparison with scientific deduction, the point (plot) of a story can be either implicit or explicit,
and causal links between events in the story are interpretative, rendering narrative a looser inferential framework.

We explore how storytelling (the process) and stories (or narratives) involving scientists can make sense of environmental crises, where conditions change rapidly and natural, social, and scientific systems collide.  We use the example of the Soufrière Hills volcanic eruption (Montserrat), and scientists' experiences of the events during that time. We used 37 stories gathered from seven semi-structured interviews and one group interview (5 scientists). We wanted to understand whether
these stories generate or highlight knowledge and information that do not necessarily appear in more conventional scientific literatures produced in relation to  environmental crisis, and how that knowledge explicitly or implicitly shapes future actions and views.

Through our analysis of the value these stories brings to volcanic risk reduction we argue that **scientists** create and transmit important knowledge about risk reduction through the stories they tell one another. In our example  storytelling and stories
are used in several ways: (1) evidencing the value of robust long-term monitoring strategies during crises; (2) exploring the current limits of scientific rationality, and the role of instinct in a crisis and (3) the examination of the interactions and outcomes of wide-ranging drivers of population risk. More broadly these stories allowed for the emotional intensity of these experiences to be acknowledged and discussed; the actions and outcomes of the storytelling are important. This is not about the 'story' of research findings but the sharing of experience and important knowledge about how to manage and cope with
volcanic crises.  We suggest that storytelling frameworks could be better harnessed in both volcanic and other contexts

## 1 Introduction

*'we gain explanatory power by thinking of natural and social orders as being produced together.' (Jasanoff, 2004*)

In many areas of the environmental sciences, the spatial and temporal scales over which natural processes operate translates into a state in which, '*the world is not a solid continent of facts sprinkled by a few lakes of uncertainties, but a vast ocean of*

*uncertainties speckled by a few islands of calibrated and stabilized forms.*' Latour (2005**).** This is particularly true during the moments of an environmental crisis, where unforeseen (or unmitigated) human actions combine with natural systems or processes to create unstable or dangerous conditions that require warnings and action on short timescales. In these circumstances the ambiguity of scientific knowledge, and consequent incapacity to accurately forecast change on a relevant timescale means that the inferences, caveats and ambivalence derived from narrative is an attractive way to offer the best sense to be made, rather than definitive explanations or singular possible outcomes of cause and effect.

However, storytelling in the context of science, scientists and scientific knowledge enjoys a troubled if productive relationship. At first glance, a distinction can be made between scientific and narrative descriptions of the world. Definitions of the scientific world assert that it is verifiable and reproducible by objective observation, experiment or model. The narrative world instead charts unique paths through sequences of events, controlled, and therefore coloured by the choice and emphasis of the narrator (Labov and Waletzky, 1967). However, scientific understanding is rarely complete and the construction of scientific truths also necessarily involves the distillation of key variables, or the jettisoning of observations and facts not central to the 'plot' or hypothesis to be tested (Padian, 2018). Scientific narratives can also share collectively shaped plot morphologies in similar ways to other types of 'story' (Hoffman, 2014). Nonetheless, a critical distinction often remains: while narratives seek to merely imply or convey meaning for a possibly unique lived reality, the central goal of a scientific narrative is to convince the reader of the universality of the proffered evidence (Dahlstrom, 2021). Narrative analysis of policy and processes that address environmental risk suggests that reliance on scientific-technical narratives of environmental risk embed conservative approaches that impede transformation in the face of long-term challenges (Borie et al., 2019)

In the context of environmental crises there is a distinction to be made between the analysis of a narrative in its stable form (for example as a written communication or policy) and the process of constructing and sharing narratives to describe events (storytelling). Storytelling now enjoys an elevated role in conveying and communicating scientific information as it also widens opportunities to share meaning and knowledge (Dahlstrom, 2014; Elshafie, 2018). Using the personal and the particular enhances salience and offers the opportunity to draw in listeners who may otherwise tire of the message or messenger (Storr, 2019). For example, narratives are strongly encouraged as means to engage non-scientists in critical messages around a changing climate (Corner et al., 2018, Shepherd et al., 2018).

Thus, part of the focus of this paper is to understand how storytelling functions as a means to both share and to shape knowledge (Jasanoff, 2004, Swedlow 2012, Jackson, 2002). However, our aim here is not just to examine storytelling as a technique for increased comprehension and engagement (Dahlstrom, 2014) but to understand how it serves all participants, in this instance those who tell the stories themselves: the scientists.

This is because understanding and making sense of the intersection between the natural environment and human cultures generates profound challenges for those also engaged in minimising impacts. Complexities and heterogeneities across multiple hazard and social systems can obscure pathways to the most effective collection or use of scientific information. The way in which scientists navigate and then make sense of these situations is under-explored, particularly for the role it

plays in how they make sense of the crisis, both in the moment and in governing their future actions and understanding. In most situations of environmental risk it is acknowledged that hazard scientists have some power (in that they both implicitly and explicitly hold a seat at the decision-making table during a crisis) but with that power comes an obligation to anticipate what might happen next or to outline 'what to do'. So their lived experience of the crisis may not feel powerful to them or reflect external perceptions of tensions that exist. Official articulations of hazard and risk, such as peer-reviewed papers, only cover part of the insights into crisis response, conforming to expectations around disciplinary boundaries or norms. However, we argue it is in the space where natural and human systems collide that valuable lessons for coping with rapid environmental change lie; and that the analysis of sense-making through storytelling has much to offer, partly because of the fluidity of interpretation implicit in their structure, which matches the uncertain and evolving understandings. This is complementary to the conceptualisation of 'storylines' emerging in the climate science community (Shepherd et al., 2018) where narrative use is highlighted as a means to explore plausible future climates, or use past events as demonstrators of future scenarios.

We develop this further, and use the specific example of a volcanic crisis, and scientists' stories of the events that surrounded a series of eruptions during that time to explore the value of narratives to sense-making through the lens of an environmental crisis, where decision-making is often dependent on rapidly changing and ambiguous scientific information, on short time-scales. We wanted to understand whether stories told of such moments generate or highlight knowledge and information that do not necessarily appear in more conventional scientific literatures or learnings from an environmental crisis, and how that 'tacit' knowledge shapes future actions and views. . We wanted to understand the various ways that the recounting and recalling of events during volcanic eruptions happen and how this shaped the understanding of the storytellers. In this context we are dealing with a risk 'system' analogous to many other human-natural risk systems particularly those where the timescale for action is less than the timescale over which the risk and its implications can be fully understood.

We show that the stories told by scientists of volcanic crises increase their explanatory powers, providing an important means to sense how natural and social 'orders' combine to create volcanic risk during eruptive crises. Importantly, scientists acknowledge the pervasive influence of uncertainty in these moments. Because narratives are functionally rooted in experience and the violation of expectations that accompanies or drives uncertain situations (Hyvärinen, 2016), they are well suited to describing and analysing crisis events. Further, decision-making or actions under conditions of uncertainty, can create discomfort for scientists when responding to these crises and these emotions are not only strong drivers for storytelling but provide outlets for the emotions associated with these experiences. Consequently, we demonstrate that in addition to the knowledge in the content of the narrative, the action of storytelling, and its affect are important as a means to process events (Jackson, 2002; Goodwin, 2015).

Through our exploration of the value the information in these stories brings to volcanic risk reduction we argue that **scientists** create and transmit important knowledge about risk reduction through the stories they tell *one another*. This is not

about the 'story' of research findings but the sharing of experience and important knowledge about how to manage and cope with volcanic crises and we suggest that this type of knowledge could be better harnessed in both volcanic and other contexts.

## 2. Our methodology, framework and rationale

### 2.1 Interview framework and Hazard Context

In the context of this paper we follow the rationale of Polletta et al., (2011) and treat story and narrative as interchangeable terms that convey the act of descriptive sequencing of events in order to make a point. Our methodological framework was shaped by our desire to understand how the stories that scientists tell during and after a crisis can help to shape ideas and 110 knowledge around them: the thematic analysis was thus particularly shaped by theoretical understandings of the drivers of risk during volcanic crises. Crucially, the point of a story can be either implicit or explicit, and causal links between events in the story are revealed via the plot, rendering narrative a looser inferential framework than scientific deduction (Padian, 2018). Furthermore, for scientists who respond to a crisis, sense needs to be made of the situations that arise, and these can have a profound emotional as well as scientific resonance. A rich literature explores emotional responses to disasters and the 115 control it exerts on memory, decision-making and subsequent behaviour (e.g. Walshe et al., 2020; Monteil et al, 2020). However, emotional impacts on scientists themselves are relatively underexplored, nor the affect that the emotional intensity of the experience has on the value of this experience to the involved scientist.

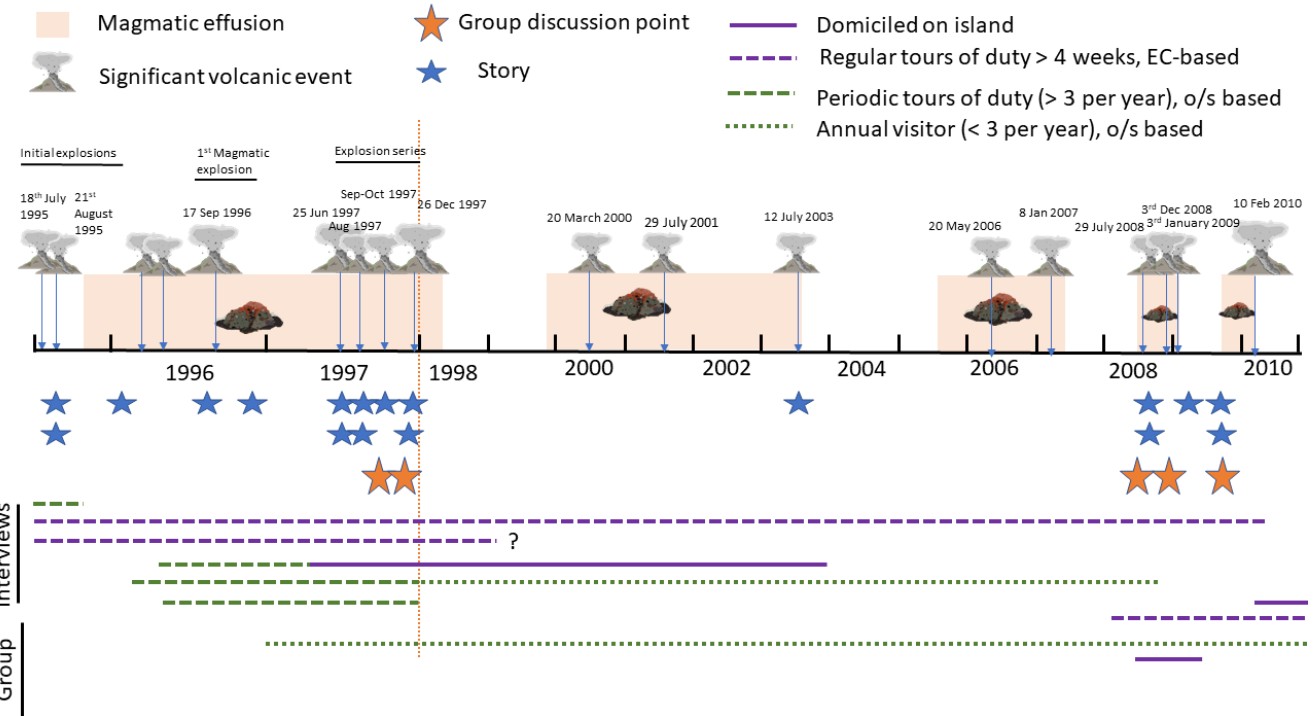

Figure 1. *Timeline of stories told. Shown here are also significant volcanic events during the 1995-2010 eruption of Soufrière Hills Volcano and relationship with the presence, and nature of work of storytellers. Vertical orange-dashed line indicates a break in the temporal scale at the end of 1997. Blue arrows show more precise location of some of the event. Significant events include major dome collapses, and explosive activity, set against periods of magmatic extrusion and quiescence. Timings of these events are taken from Wadge et al., 2014.* EC = Eastern Caribbean., o/s = overseas

To create data relevant to the aims of our study, we conducted 7 semi-structured interviews with scientists involved with the volcanic crises of the Soufrière Hills Volcano (SHV) on Montserrat (1995-2010, Figure 1). This is an exceptionally well studied and understood eruptive crisis (e.g. Druitt and Kokelaar, 2002; Wadge et al, 2014), with a wide variety of events and analyses on which to draw (scientific narratives and socio-scientific analysis).  Crucially volcanic eruptions such as these are analogous to other examples of rapid environmental change, where there is an oscillation between moments (hours, days or weeks) of acute crisis, interspersed with longer periods where threats are still present but less immediately visible to all populations. In the case of this time period on Montserrat, acute crises were presented when SHV generated or strongly threatened impactful pyroclastic density currents or explosions (Figure 1) with periods of enhanced background risk presented by some periods of apparently benign magmatic effusion (passive extrusion of volcanic material into the volcanic crater) or when unseen or felt subsurface geophysical signals indicated an increasing likelihood of further destructive activity. Combined with current long-term knowledge of these systems (hazard analysis), the collection of monitoring data

creates the possibility of identifying and anticipating changing conditions over hours and days, and to describe the range of likely behaviours: but not to 'forecast' with any certainty. Thus, scientists involved in these crises can contribute knowledge salient to decision-making but only with defined uncertainties in the best cases, and in many instances the interpretation of monitoring data can be ambiguous and difficult.

In collecting the scientists' narratives around an environmental crisis such as this, the choice of events, the nature of their description, their sequencing and plotting are all then important to understand. And, as storytelling is inherently a political and social act (Jackson, 2002; Polletta et al., 2011; Koch et al., 2021); understanding the process of storytelling, is as important as the content of the stories themselves. Finally, we needed to be able to infer how the stories themselves and how the rationalisations behind them act to shape knowledge.

Thus, during the course of the interview we gave the scientists free rein to tell any three stories of their choosing that related to the volcanic crisis, and asked them to tell it as closely as possible to how they might normally do. Three stories widened opportunities for variation in focus and mode of telling. Participants were told in advance we were looking for these stories, to allow them time to prepare if they wished. On completion of their storytelling we also asked questions to add further contextual information about their story, if needed. In particular, we asked them how, where, when and why they more normally would tell these stories; the points they were aspiring to make (implicitly or explicitly); and whether the story changed in form, or shaped knowledge.

Our interviewees represented different backgrounds and career stages) present at different and sometimes overlapping times (Figure 1) during the Soufriere Hills eruption. Our complete interview protocol can be found in the Supplementary Materials.

Our research team is two volcanologists, and a social scientist, albeit each with considerable experience of working across disciplinary boundaries, particularly in the context of volcanic risk (e.g. Armijos et al., 2017; Barclay et al., 2019, 2022). We chose to have the two volcanologists conduct the interviews, to simulate as closely as possible a situation where scientists share stories with one another. To supplement the individual interviews, we also conducted one informal group conversation with a further five individuals present, all of whom had experienced volcanic crises, not all at SHV (Fig. 1). This was an opportunity to compare distinctions between one-on-one conversations and a situation resembling group discussion. During the group interview, we only asked for clarifications relating to context, and allowed conversation to freely explore the meaning or point of each story and to make comparisons across experiences. Each of these encounters was transcribed in full for thematic coding.

## 2.1 Analytical Framework

As well as using the story and respondent answers to questions to generate basic descriptions of the story, its timing, and the broad theme or expressed point or 'plot' of the story (Table 1 and Figure 1), we also identified several themes using

structures, devices and contexts of storytelling (e.g. Jackson, 2002, Storr, 2020) to provide a thematic coding ('*a*' in Figure 2).

This was further developed to integrate themes relevant to the description of natural risks in general, and volcanic risk in particular. For example, acknowledging the importance of uncertainty and ambiguity in interpreting information (Stirling, 2010, Sword-Daniels et al., 2018); the dynamism (e.g. Brown et al., 2015) and politicization (e.g. Donovan 2021a and b,) of

175 volcanic risk and the challenges of evacuation (Barclay et al., 2019). So, in addition to the narrative descriptions of the stories ('turning points') a further three interdisciplinary themes dissected the content of the natural socio-scientific knowledge embedded in the storytelling and the purpose behind the storytelling. These were: (*b*) improved understanding of risk drivers and the consequences (*c*) counterfactual analysis which attributes cause and effect between actions and outcomes or what would have happened had events unfolded slightly differently; and (d) how knowledge is currently used and shared

('propagation'), and how it has shaped attitudes and actions, both for the knowledge it contained and the sense it creates in the telling.   A brief description of our themes and sub-themes and their relation to cognate literature is introduced in Figure 2, and in the Supplementary Material we illustrate the stories further with anonymized data from the 'volcanic turning points' sub-theme (see results).

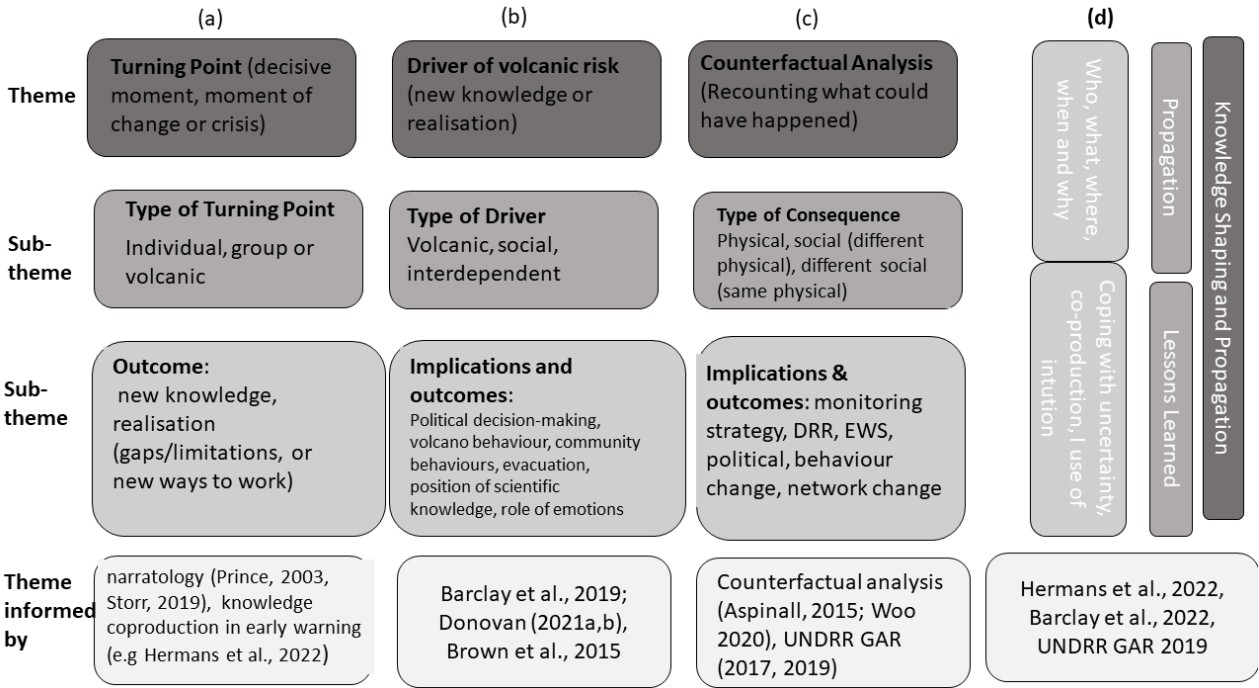

*Figure 2: Diagram of themes. For a fuller description of themes and the framing literature please see Supplementary Materials*

## 3. RESULTS

### 3.1 Storytelling context and focus

In total our seven interviews and our focus group generated 37 stories (Fig 1 and Table 1). In analyzing the focus group conversation we also sub-divided those stories told by individuals within the group and more flowing moments of conversation where stories were told to exchange ideas or illustrate points ('Discussion point'). In the focus group, just four stories were told as a standalone story before the group moved to a conversational mode where stories were used to enhance debate (creating a further 10 'stories' many of them secondhand or comparative). The stories told by individuals, and those initially told by the group often represented a 'turning point' or moment at which a decisive change in a situation occurs. However, that moment was not always directly driven by new volcanic phenomena, or volcanism new to the storyteller (8 storytellers, 15 stories, see Appendix 3) but about the relationships between the volcano and social situation (all storytellers) at multiple scales:

*'there was very much a feeling then of the island is so depressed and squeezed that, yeah, it was quite an upsetting time, but then it was a time that you feel, well, it's a moment in your life that you're pleased to have been there to experience it but you don't want to see it again.*

***Story 4***

These stories also revealed the emotional intensity of scientific involvement in a volcanic event. Participants often described the emotions associated with these moments or turning points, both for them and for others involved. In the case of the volcanic turning points these emotions involved interest, excitement, dedication, and surprise as well as some level of fear and anxiety, particularly around what might happen next. For those turning points with a wider focus, storytellers also discussed feelings of inadequacy, learning, confidence and humility around their capacity to offer solutions and commonly offered lessons they had internalised and shared, from the practical to the philosophical.

'*if the volcano is showing signs of activity, when you go home you make sure and put gear inside your truck! Gas mask and helmet.*' **Story 13**

'*I tell often to young scientists and to people to make them realise that in a hazardous situation you really have to have your head on and you have to be prepared for things that you don't really plan for*' ***Story 9***

An important dimension here was also the use of humour, which was described explicitly by five of our storytellers as a vehicle to enhance the value or saliency of their story, but also in the moment, to diffuse tension, and both implicitly and

explicitly to illustrate the humanity (and limitations) of the storyteller. It was also used by the group during their discussions to diffuse tensions around differing points of view, and was a feature of mode of retelling in their stories. As one storyteller remarked about humour:

'*those little personal hooks, I think they're a big part in the whole sort of web of it because you're not there as a passive observer*'                    ***Story Reflection***

### 3.2 Story Plots

Basic plot types are summarised in Table 1. A critical feature of many of these plots were their descriptions of working with uncertainties (both human and volcanic) and the repercussions of that, particularly inadequacies in decision-making in the face of these uncertainties. Certainly in telling these stories, the volcanologists largely moved far from the volcano as a creator of volcanic hazard alone and uncovered these crises as moments for reflection on how risk is considered (Bankoff, 2021). From our thematic analysis, exploration of uncertainty was also closely associated with descriptions of the range and temporal and spatial variability of volcanic behaviour (like Soufrière Hills Volcano, volcanic eruptions can be protracted with wide variations in the human impact of activity during that time). In these instances the conclusion of the story often involved a counterfactual analysis (Figure 2) of 'what might or could have been' or an improved understanding of the social and physical consequences of the volcanic behaviour. The lessons subsequently drawn by the storyteller, involved  their reflection on what they would or had done better the next time.

| Story # | Plot | Experience |
|---|---|---|
| 1* | Scientific opportunity is difficult to square with social consequences | Witnessing early phreatic explosion, August 1995 |
| 2 | Scientists have to use their instinct in acute situations | Response to first cold flow into Plymouth, August 1995 |
| 3* | Scientific knowledge has limits, increased by crisis observations (other) | Large explosive eruption, Redoubt volcano, 1989 |
| 4 | Unexpected behaviour and volcanic uncertainty is hard to deal with (other) | Unexpected eruption, Mt. St. Helens, 2004 |
| 5* | Larger than average events generate emotions and expectations | Events, decision-making & impacts of fatal pyroclastic flows, 25th June 1997 |
| 6 | Volcanic uncertainty has a profound influence on managing the risk | Communicating and anticipating eruption end, August 2009 |
| 7 | Politics of risk management as volcanic activity escalates | Communicating escalating volcanic |

| | | activity, December 2009 |
|---|---|---|
| 8* | Volcanoes generate unexpected events and you need to be ready | Overnight explosion (first magmatic explosion) 17/9/1996 |
| 9* | Scientists have to use their instinct in acute situations, risk perception | Response to first cold flow into Plymouth, August 1995. |
| 10* | Volcanic uncertainty has a profound influence on managing the risk | Responding to explosive events in September 1997 |
| 11 | Sustaining volcanic monitoring creates risky personal choices for scientists | Need to sample new lava dome in January 1996 |
| 12* | Volcanic variability requires wide range of preparedness | Responding to a volcanic explosion, March 2003 |
| 13 | Volcanic variability requires wide range of preparedness | Responding to a volcanic explosion, August 2008 |
| 14 | One needs to be ready for unexpected events | Maintaining monitoring systems near to volcano |
| 15* | Alerting authorities to respond when likelihood of eruption uncertain | Unrest into eruption 1992-1995 |
| 16 | Whose scientific voice gets heard when interpretation is uncertain | Unrest into eruption 1992-1995 |
| 17* | Scientists, instinct and luck in uncertain situations (other) | Rapid unrest into explosion La Soufrière, St Vincent (1979) |
| 18* | Emotional intensity of experiences during monitoring, and communication of likely escalation of activity | Events, decision-making & impacts of fatal pyroclastic flows, 25th June 1997 |
| 19* | Continuing volcanic activity has profound social and scientific consequences | Impacts of intense activity in August and September 1997 |
| 20 | Politics of risk management, and aid assistance | Large dome collapse and flows, July 2003 |
| 21 | Small changes in flow (or eruption) pathways generate disproportionately greater impact ('luck') | Escalating activity in June 1997 |
| 22 | Politics of risk management | Impacts of intense activity in August and September 1997 |
| 23* | Behaviour variability: conveying unexpected or larger behaviour | Anticipating major volcanic edifice collapse in 1996/1997 |
| 24 | Political influences and scientific decision-making (science and politics) | Response to intense volcanic activity in late 1997 |
| 25* | Uncertainty from variance in behaviour (other) | Unexpectedly large explosion at Bromo volcano |

| 26* | Uncertainty from variance in behaviour | Risk variance at one location between April and December 1997 |
|---|---|---|
| 27 | Scientific preparedness for volcanic behaviour | Major edifice collapse December 1997 |
| 28 | Uncertain volcanic behaviour and decision-making | Responding to escalating volcanic activity December 2008 |
| 29 | Discussion point: decision making and communication under uncertainty | Comparison 2008 with escalating activity in 2009 |
| 30 | Discussion point: Responsibility for evacuation, complacency with patterns of eruption | General discussion of evacuation decision-making at SHV |
| 31 | Discussion Point: Respecting Risks during unexpected behaviour (other) | Exposure to sudden escalation of activity at Etna |
| 32* | Discussion Point: Unexpected cascading risks from activity (other) | Lava interaction with water tank at Etna |
| 33 | Discussion point: evacuation and risk | Comparison across settings of risk management procedure |
| 34 | Discussion Point: 'Luck' vs good risk management, scientific objectivity and instinct (SHV/other) | Comparison of sudden escalation at Stromboli and SHV |
| 35 | Discussion point: Sudden acceleration in behaviour and risk management (SHV/other) | Challenges of evacuation, livelihoods and islands around anticipating escalation (SHV, Agung, Ambae) |
| 36 | Discussion Point: Uncertainty: Expecting the unexpected ('plan for failure') (SHV/other) | Anticipating activity (St Vincent, Stromboli, SHV, NZ) |
| 37 | Discussion Point: Eruption size and proximity (SHV/other) | High consequences of proximity and understanding type of eruption (Galeras, Colima, NZ, La Reunion, Kagoshima) |

*Table 3 Summary plots and knowledge boundaries of the 37 stories and discussion points. (other) in brackets denotes that story or discussion contained elements from a different eruption. \* denotes a story where the 'volcanic turning point' is illustrated in Appendix 2. SHV = Soufriere Hills Volcano. Themes developed in the discussion are underlined. NZ = response to an eruption in New Zealand. All other volcanoes are referenced by name. Pyroclastic flows are the dense component of hot flows of lava, blocks ash and gas generated at volcanoes like SHV. These can move at great speeds and are usually fatal to those caught in their path.*

With this background many of the stories acted to demonstrate the importance of preparedness during a crisis or to reflect on the extent that instinct and emotions have a role to play in scientific decision-making. Instinct and emotion may be unfamiliar types of knowledge used within scientific norms but intersect with the concept of 'tacit knowledge' developed in the sociology of science and related fields (Polanyi, 1966). Several plots that focussed on risk rather than hazard alone

served to illustrate the influence of other risk dimensions (politics, social vulnerabilities) in decision-making processes. Again resolution (of the story) typically produced further reflection from the storyteller. This usually involved dissecting the key driver of vulnerabilities (both social and physical) during the story, and sometimes implied improvements for the future.

Consequently, during our thematic analysis we identified some 169 instances where the story represents an example of some type of new or improved knowledge developing as a result of this experience. This is demonstrated here by the breakdown in this theme of improved understandings of primarily volcanic, primarily social or interdependent drivers of volcanic risk shown in Table 2. Much of the improved knowledge described was captured during the storyteller's analysis of the interactions between the natural and social systems ('interdependent drivers', 109 instances). It is worth noting that the examples used were particularly rich as many generated knowledge improvement across more than one of the sub-themes used in our analysis.

The group discussion stories centred around the same sub-themes (Figure 2) as the individual stories. The group notably focussed on the awareness and detectability of variations in volcanic behaviour, the influence of uncertainty, and interactions with decision-makers around evacuation and warnings.

For example:

*'Whereas from me the key learning for me from the whole process – especially with the successful evacuation, probably slightly more on the social aspects of it. After the evacuation was successful I was pretty much …...I had a lot of compliments from a lot of people.. but the one thing... the first thing that came to my mind was ' yeah, what would you think... if we had had an evacuation and nothing had happened… for a week… two weeks…three weeks….'*

*Storytelling group discussion*

As this discussion escalated participants drew on other volcano 'stories' (both via direct experience and those passed on to them, Table 1) to illustrate their points, using these examples to shape each other's thinking.

*'There was an element of it being a fortunate thing. And the real point is that there was this real, sudden acceleration outside the paradigms of which we had already been working. And for me. Having all those experiences, when I see what has been happening here. When I saw exactly the same thing you have been [describing]'* **Storytelling group discussions**

| Knowledge Improvement     Sub-theme | No. |
|---|---|
| **Volcanic Drivers** | 15 |

| | | |
|---|---|---|
| | Signs and Signals (monitoring, time and scale over which they occur, repeatability or generalisability of activity | 11 |
| | Realisation about impacts from activity, including its uncertainty | 17 |
| **Social Drivers** | | 7 |
| | Political decision-making (including outcomes of and influences on ) | 14 |
| | Community behaviour | 3 |
| **Interdependent Drivers** | | 47 |
| | Role and position of scientific knowledge (its influence and its power and respect and confidence in that knowledge) | 19 |
| | Evacuations and warnings | 23 |
| | Emotions role in decision-making | 8 |
| | Cascading Impacts | 5 |

**Table 2:** *Occurrence of 'Knowledge Improvement' Themes Detected in Stories and their description. No. shows the number of times this sub-theme was identified in stories. Where instances are ascribed to the main subtheme these are examples without clear alignment with any of the identified sub-themes, either because they are less recurrent examples,*
*or because the example includes several sub-themes.*

These moments of story swopping and analysis perhaps best represents the point at which storytelling interviews got the closest to the informal conversations that volcanologists might share with one another. However, across this group and in our individual interviews very few identified storytelling as a space or moment where they consciously learned from each other or that helped them to cope with their experiences (in response to questions about how, why and where they tell these
285 stories). On further probing most agreed that they at least re-told them within their own institute (particularly to colleagues with less direct experience), during official professional presentations, and (where applicable) while teaching at undergraduate level, or as a tool for public engagement. However, this is not universally the case, one storyteller admitted that this was the first time they had spoken one of their stories out loud to anyone. In educational settings storytellers recognised the value of the story as a vehicle for conveying complex inter-related drivers of volcanic risk, but in a peer-to-
290 peer context volcanologists were more likely to identify sharing these stories as sources of entertainment and as a means to

build relationships. However, when prompted, most storytellers agreed that there was knowledge embedded in these stories that was not available conventionally and several spontaneously offered the names and instances of peers and senior colleagues whose stories had taught them valuable lessons.

## 3. DISCUSSION

### 3.1 Volcanic storytelling in context

There is a strong (> 150 year) tradition of integrating differing narrative accounts of eruptions within volcanology (see e.g., Krakatau Symonds et al., 1888 ; Soufrière St. Vincent Anderson and Flett, 1903, Fiske, 1979) and a recognised flourishing of understanding associated with their publication. These typically bring together official records, existing scientific datasets and 'unofficial' accounts contributed by or solicited from other individuals. Several integrated accounts that also include personal views, experiences and insights have been written as 'popular science' books (e.g. Patullo, 2000; Winchester, 2005 ). These are all widely referred to and used in the academic literature as sources of descriptions of volcanic impacts or phenomena not otherwise available, implicitly representing knowledge or understanding that would otherwise be unavailable.

More recently single or multi-author monographs have been superceded by multi-focal 'special volumes' which often contain a similar range of sources but divided by discipline. The Soufrière Hills eruption has been well served by both formal and popular accounts of events (e.g. Druitt and Kokelaar, 2002. Wadge et al., 2014, Patullo, 2000 ),  but most of our storytellers identified that their stories were not fully represented in this literature, and a few others that they were only partially covered in specific cases.  This assertion is consistent with the richness of insights developed via the analysis of literature arising from the crisis (Donovan et al., 2011) and the recent use of narratives in examining 5 perspectives on risk for Mount Mayon (Bankoff, 2021).

Thus in published work to date there is insights missing, as one storyteller remarked '*A publication is your analysis and science bit, and there is that more sort of on the ground experience that you accumulate. But you really accumulate that individually... and, you know, you learn and gain a certain amount from other people's experience and you should do where you can, but you can't just distil what [Scientist] has got and download it into one of us [scientists]*'. Further, many storytellers acknowledged the intensity of the experience and the need to share and process these experiences  '*emotional experiences are still a very basic, fundamental, human response and they need to be addressed for people to be whole as they go forward and ... there is the adage that those who do not remember their history are doomed to repeat it, and so if these stories are not told, they're not somewhere, it can happen again*'.

Our analysis clearly shows that storytelling has value in helping scientists to describe and make sense of societal interactions at the limits of their knowledge and where actions are not and cannot be solely influenced by scientific advice when the uncertainty around possible outcomes is high.  Uncertainty, and the lived experience of coping with and responding to scientific uncertainty lies at the heart of much of this discourse. The crossing of borders between epistemic and instinctive

knowledge and the need for the scientists to live within the uncertain environment they are experiencing means that much of the uncertainty here is 'embodied uncertainty', which is the subjective experience of uncertainty for those living at risk (Sword-Daniels et al., 2018). Storytelling offers a means by which this can be better understood and represented.

So, here we identify core lessons embedded in these stories that augment understandings of volcanic uncertainty and risk. We then also reflect on their collective value in understanding the experience of living with uncertain situations and in responding to environmental crises scientists as well as the value to the storytellers themselves in acknowledging and processing the impact of witnessing an eruption, and its role in shaping their future actions. Finally, we reflect on other crisis contexts where storytelling as exemplified here could be a useful tool.

### 3.2 Volcanological lessons in uncertainty and risk: Preparing for the unexpected, the prolonged and the variable

The stories told about the SHV eruption by many scientists reflect an eruptive situation with: wide-ranging volcanic behaviour; the constant need to reinforce and update the monitoring network as the eruption progressed (reactive rather than proactive response); balancing inputs from both local and external volcanologists, and resource constraints. No one volcanic crisis is typical but these parameters are representative of resource-constrained settings (Joseph et al., 2022). Responses and reflections on these challenges emerged in our stories and collectively provide convincing qualitative evidence for the value of investment in good monitoring networks in advance of coping with a crisis.

*'ideally, you'd have everything processed and analysed either on the fly or shortly thereafter. But at least so that in retrospect, you've captured the situation and you can utilise the data... I mean the intense times, it's like a bonfire night or something. It's a big, dramatic time and that's what appears in the news or whatever it is. I mean, most of the work at the observatory isn't that, most is... I'm not saying mundane but there's a lot of day-to-day stuff and things that go on and on.. **but an awful lot about it is having that strength in what you do**. And there's a tendency to tell stories about .. dramatic bits of volcanic activity. But they're not the equilibrium'*      ***Story Reflection***

Thinking through the challenges in advance and essentially preparing to 'expect the unexpected' were also a common feature of interviews along with the impacts of coping and dealing with intense escalations.

*'for me the key learning there was that even with the best instrumentation you can have, the volcano behaviour can change through time, and that was a big lesson...the real point is that there was this real, sudden acceleration outside the paradigms of which we had already been working'*      ***Group Discussion***

*'and, it was quite interesting to see very experienced volcanologists looking completely helpless thinking about the challenges of these little eruptions.'*                                                    ***Group Discussion***

The prolonged, variable and uncertain volcanic behaviour also formed core plot points for several of our stories (Table 1) and the issue of crossing between epistemic or knowable features of activity into situations where scientists were relying or acting on instinct was discussed, even if it was uncomfortable to do so (see below). Frequently storytellers had benefited from their own informal counterfactual analysis of what could otherwise have happened in a variety of different situations.

*'I never had the instinct, which I probably should have, to run.'*                                    ***Story 2***

**3.3 Decision-making under conditions of uncertainty: the limits of 'rationality'**

Embedded in monitoring-focused stories were of course lessons in 'what to expect' during an eruptive crisis and to be ready for a wide range of demands, not all of them scientific

*'you have to see many things at the same time. So you have to be at one point alert to the public, you have to be making sure the observatory's running, you have to make sure that the staff were doing what they were supposed to, you have to fix the*
*generator, you have to deal with people who are coming there'*                                    ***Story 8***

Reflecting on wider decision-making under uncertainty, for example in the context of changing alert levels, evacuations or even personal risk, the acknowledgement of the difficult boundary between quantifiable or even knowable risks and instinctive decisions were important topics that are otherwise difficult to unearth in the literature. Particularly, the experience of the exceedance of scientific 'boundaries' either by involvement in political decisions, or using instinct (or experience) to
understand when risk is increasing, or simply articulating the conditions under which scientists become less certain

*'we tend to be very rational and quantitative as possible. So the thing is I do not recall very many public meetings in various countries where someone would not come at the end and say 'what would you do' – basically just reverting to the emotional rather than just based on facts. And you see this over and over and over again.'*

*'....someone has to make a decision based on an increasing level of uncertainty... There are more differences of opinion for anyone who is in charge'*

*'there are also different ways, different reasons why you may want to propose or recommend an evacuation. One of them is that you have evidence that something is about to go pear-shaped, and the other one is simply because your uncertainty is*
*increasing'*                                                                                ***Group Discussions***

The opening up of these conversations rather than closing them down created insights into how these situations, their risk and embodied uncertainty were negotiated between the monitoring scientists, managers of risk and populations at risk.

Scientists recognised the variance in power between actors in a crisis situation, and the implicit lack of tolerance for uncertainty from them that this scientific power brought. Apparently 'getting it wrong' then came with high stakes in terms of trust.

' *[your advice is] going to be questioned like never before because evacuating people from their homes is such a hugely emotive problem….You've got to be absolutely, 110%... and I was absolutely confident and that was the only way I could kind of cope with the stress of the situation because I was absolutely convinced at the time that what I was doing was the right thing to be doing.'*                                                                                    **Story 7**

In a similar vein these stories and importantly the counterfactual analysis of associated 'near misses' generated opportunities to learn from failures, not only to improve for the next time but to maintain perspective on the limits of a science like volcanology when applied in a crisis setting in that moment.

*'A: I think talking about failures , talking about those near misses. It really depends on the circumstances. When we put ourselves in those situations to a certain extent its probably something we're not too keen to talk about. In cases where we would probably do exactly the same thing, but its just freak things that happen. I don't like talking about it.*

*B: but do you think you should be talking about it?*

*A: I think so, I think its better. Because otherwise I think we entertain the idea that we can forecast eruptions.'*

To prepare for these moments volcanologists have recently developed a range of tools and simulations to prepare for the conditions of uncertainty, through crisis simulation (Bretton, 2018, Ang et al., 2020) and the development and evaluation of careful communication protocols and guidelines (IAVCEI, 1999, 2018). We demonstrate here that these stories and their analysis are powerful further tool to understand critical drivers of risk, and in so doing better evidence how to prepare for future events.

**3.4 The value to the scientists and science of telling these stories**

At its most basic level telling these stories offers a powerful way to communicate risk and the moment of a volcanic eruptions in a way that is more memorable (Storr, 2019 ) and that enhances salience and offers meaning (Dahlstrom, 2014).

*'I try and tell the students…to convey that feel of it, you know, 'This is just volcanic pandemonium!'*                    ***Story ref***

This process of storytelling also offers benefits to the storytellers, the introduction of feelings, senses and emotions into the story allows the teller to process and share what has happened to them, an element essential to recovery and processing of being involved in a disaster (Cox et al., 2017).

*'I think one of the things that was not done that should have been done, there should have been a period after the events of that day for people to talk and ... so that they could have that kind of closure'* **Story Reflection**

Our thematic analysis of the content of the stories themselves provides some insights into this value, which could be developed even further by drawing more deeply on narrative analysis (e.g. DeFina and GeorgakoPoulou 2015). This demonstrates that the value of the narrative is shaped by more than the content alone, the interactions between

'storytellers' and their audiences, and the physical and unspoken vocabularies create and shape further knowledge as it is used (Goodwin, 2015). As an illustration of this potential, we noted from our own experience, that during our informal group conversation the storytelling process most closely resembled natural conversations between scientists, it is noteworthy here that the stories escalated into nine further discussion points of direct relevance to responding to volcanic risk (Table 1), involving the sharing and shaping of each other's ideas. However, the fact that most of our storytellers

identified the value of swopping and sharing and stories with one another primarily as a means of entertainment or social bonding shows that their inherent value is currently underestimated within our community. New and further work could develop this important avenue by using narrative analysis to create a framework for developing storytelling as a means to understand volcanic risk.

Further, monitoring agencies in some settings can offer professional and individual counselling and psychological support

in the wake of responding to crises that become disasters. This is perhaps a minimum standard but the reflection in the stories told here, where scientists are crossing the boundaries and limits of their scientific capacity demonstrate that there is also value in open sharing of stories between one another to enable them to make sense of the professional situation, and to acknowledge new ways of coping with those uncertainties.

### 3.5 Narratives and uncertain crisis contexts

In the case of volcanic eruptions, we have demonstrated that narratives have value to the storytellers for the post-hoc rationalisation of complex, changing and uncertain situations. By analysing series of stories common themes emerge that provide insights into not only the most important challenges for scientific monitoring but into the situations that drive risk-taking behaviour by scientists, or that impact effective decision-making by individuals or institutions involved in the crises.

The stories told of crisis moments could be an important additional tool for other situations where decision-making is

constrained by the uncertain temporal and spatial timescales of impact created by uncertain and dynamic hazard behaviour, or the cascading impacts between social, political and cultural landscapes and the initiating phenomena. When considered in this way there are direct parallels between volcanic crises and the decision-making challenges associated with, for example,

the COVID-19 pandemic (Berger et al;. 2021) ; the conveyance  and analysis of the uncertainty associated with different local impacts and scenarios emerging from climate change  (Kemp et al., 2022).  Shepherd et al., (2018) in particular have

demonstrated 'storylines' are a valuable tool in moving from the characterisation of the drivers of climate change to the creation of actionable advice for the future, particularly where there are multiple interacting variables that generate varying uncertainties. They recognise the representation of uncertainty as a hindrance to long-term decision making. Here, we demonstrate the value of this approach in understanding the short-time scale of crises (analogous to the occurrence of the extreme weather events made more likely through climate change).

A critical challenge for scientists in all these areas is isolating evidence for actions that will improve awareness and preparedness, and to generate shared understandings that contribute to improved responses during the crisis moment. We have also shown that scientific storytelling not only evidences or helps to imagine these complex social-natural situations but that it generates new tacit understandings of the drivers of risk, that can be made explicit through analysis of this type. Similarly, the vehicle for emotional processing of crisis situations offered by storytelling is relevant in these examples too.

As more scientific communities are required to respond to the short-term extreme events associated with a changing climate the framework we provide here could provide the basis for further exploration of these situations, to process these issues, generate new knowledge and improve response and action.

*4. CONCLUSIONS*

We have presented evidence that there is a strong inherent value in stories told by scientists for the rationalization of their

experiences of complex, uncertain situations, both for the audience and for the 'storyteller' themselves. We show that inferences about cause and effect emerge, centred on the embodied uncertainty inherent in this type of situation. These allow the sharing and analysis of how volcanic risk is negotiated and decisions-made. There are also benefits to the storyteller to help them make sense of the situation they have witnessed and in introducing and acknowledging the role of emotions and feelings in dealing with these situations. The process of re-telling the story of their experiences creates new knowledge and

value to them.

 Further analyses of these stories or the creation of storytelling opportunities would not only be of value in volcanic crises but in other situations too. This methodology has the potential to yield even further insights if deepened to include further dimensions of narrative analysis to include the collection of details of the interactions during the telling of these stories,  as well as more nuanced influences of the languages of storytelling including the genres or modes of storytelling and their

impact on content relevant to understanding uncertain crisis situations.   Nonetheless, the analytical framework that we have used here could be adapted to other situations or explicitly used to identify common problems or solution in responding to future environmental crises under conditions of uncertainty

## Data Availability

Due to the nature of the interview materials and agreements with interviewees anonymised interview data are only available on request from the main author. Details of coding used are found in the Supplementary Materials.

## Author Contributions:

Development and conceptualisation was shared amongst JB, RR, TA with investigation conducted by JB and RR and formal analysis by JB with support from RR and TA. JB led the writing with contributions from all authors to the original and subsequent drafts

## Competing Interests

The authors declare they have no conflict of interest

## Acknowledgements

This work was carried out with the time created by a Royal Society APEX Award (APX/R1/180094) to JB and the time and resource from the UKRI AHRC Follow-on Award to JB,TA and RR ('Disaster Passed' AH/S009000/1 ). We would like to thank all the scientific storytellers for the generous sharing of their stories and time, many of them providing thoughtful comments on this manuscript too. Two reviewers, Matti Hyvärinen and an anonymous reviewer provided some thought-provoking perspectives that helps us to improve this manuscript and we would like to thank Wendy McMahon for early conversations on narrative analysis.

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

**Data Availability**

Due to the nature of the interview materials and agreements with interviewees anonymised interview data are only available on request from the main author. Details of coding used are found in the Supplementary Materials.

**Author Contributions:**

Development and conceptualisation was shared amongst JB,RR,TA with investigation conducted by JB and RR and formal
analysis by JB with support from RR and TA. JB led the writing with contributions from all authors to the original and subsequent drafts

## Competing Interests

The authors declare they have no conflict of interest


## Acknowledgements

This work was carried out with the time created by a Royal Society APEX Award (APX/R1/180094) to JB and the time and resource from the UKRI AHRC Follow-on Award to JB,TA and RR ('Disaster Passed' AH/S009000/1 ). We would like to thank all the scientific storytellers for the generous sharing of their stories and time, many of them providing thoughtful
comments on this manuscript too.