# Peer review of "Scientists as Story-tellers: the explanatory power of stories told about environmental crises Jenni Barclay[1], Richie E.A. Robertson[2], M. Teresa Armijos[3],"

_EGUsphere, 2023_

## Referee Comment (RC1)

**Scientists, uncertainty, and story-telling**

A review on Barclay, J., Robertson, R., and Armijos, M. T.: "Scientists as Story-tellers: the explanatory power of stories told about environmental crises.

Matti Hyvärinen, Research Director, Faculty of Social Sciences, Tampere University
Matti.Hyvarinen@tuni.fi

This is an excellent and inspiring article. Since Bruner (1986) there has been a great deal of theorizing about two different, if not even opposite, forms of knowledge, the *paradigmatic* (or 'scientific') and the *narrative*. Recent discussions about narrative and climate change have often led to such less than surprising conclusions that a narrative cannot fully capture such complex phenomena as global warming. Barclay et al. do not stop by wondering and stating the differences between these knowledge forms but proceed – very much in the spirit of Bruno Latour – to study the entanglement of these forms of knowing within the context of several volcanic crises. The knowledge forms may be different, even radically different, but during the time of crisis, when scholars seek scientifically justified responses, these forms of knowing get entangled in exciting and productive ways. I think this study and its setting has much to offer to narrative scholars and students of other kinds of crises.

The performed narrative analysis is as such impeccable enough but might profit from further subtlety. Words like "expectation", "surprise", and "uncertainty" recur in the text. Jerome Bruner (1990; 1991; cf. Hyvärinen 2016) has theorized narrative as a language and knowledge form whose function is to react to unexpected, surprising, and non-canonical events. He describes how cultures encode the expectations of the normal, proper, and canonical in the form of sequential scripts. The description of how the scientists were observing the gradually exploding volcanos, being insecure about whether the process follows the scientifically processed, expected course of events, or does it and how and when does it violate the expectations, reminds me closely of Bruner's discussion. To add one more layer, many recent narratologists (Fludernik 1996; Herman 2009; Caracciolo 2014) have defined narratives not merely as sequences of events but have accentuated the aspect of experiencing and emotions in the recounted storyworlds. As we read the article and the stories the scientists tell, we can see that intense feelings and disrupting experiences are strong drivers of telling.

One laudable move in the article (in 135) was when the interviewers "asked them how, where, when and why they more normally would tell these stories". This is a clever move since interviews have often been criticized for their "artificiality" (in most cases unfairly, but it is another story entirely). Now the researchers wanted to locate the "natural environments" of these stories by putting the interviewees to think about their telling habits. Of course, it is still different from the recording of the storytelling in situ...but when indeed can we be there in right time and place, with a recorder and proper permits? In any case, this move indicates that the researchers have not only collected stories as descriptions of something (say, volcanic crises) but wanted to address the "narrative realities" where the scientists already were telling these influential stories to their colleagues and students (see Gubrium & Holstein 2009).

Also, speaking about the methodology, the idea of asking three different stories sounds elegant and justified, precisely for letting the tellers to express a variety of experiences and interpretations. Of course, the employed thematic reading is not the most elegant or productive way of reading interview materials. In this study, it gives a rough and informative outline of the contents. By paying more attention to the interview interaction and the nuances of language use, this exciting material might offer chances for more thorough and comprehensive analysis. As regards the tools of more nuanced readings, I would recommend several chapters in De Fina & Georgakopoulou 2015 (esp. Parts III-IV). I hope the team will continue its cooperation with this intriguing theme!

Some details:

(40) The classical sociolinguists are William Labov (not Lebov) and Joshua Waletzky (not Waletsky). The philosopher, in (30) is Bruno Latour, not LaTour. (125) Jackson, 2002, is missing from the references.

References:

Bruner, J. (1986). *Actual Minds, Possible Worlds.* Cambridge, Ma.: Harvard University Press.
Bruner, J. (1990). *Acts of Meaning.* Cambridge: Harvard University Press.
Bruner, J. (1991). The Narrative Construction of Reality. *Critical Inquiry 18*, 1-21.
Caracciolo, M. (2014). *The Experientiality of Narrative. An enactivist approach.* Berlin: De Gruyter.
De Fina, A., & Georgakopoulou, A. (Eds.). (2015). *The Handbook of Narrative Analysis.* Chichester: Wiley Blackwell.
Fludernik, M. (1996). *Towards a 'Natural' Narratology.* London and New York: Routledge.
Gubrium, J. F., & Holstein, J. A. (2009). *Analyzing Narrative Reality.* Los Angeles, CA, London: Sage.
Herman, D. (2009). *Basic Elements of Narrative.* Malden, MA: Wiley-Blackwell.
Hyvärinen, M. (2016). Expectations and experientiality: Jerome Bruner's "canonicity and breach". *Storyworlds, 9*(1), 1--25.

---

## Author Response (AR1)

REVIEWER ONE

This is an excellent and inspiring article. Since Bruner (1986) there has been a great deal of theorizing about two different, if not even opposite, forms of knowledge, the *paradigmatic* (or 'scientific') and the *narrative*. Recent discussions about narrative and climate change have often led to such less than surprising conclusions that a narrative cannot fully capture such complex phenomena as global warming. Barclay et al. do not stop by wondering and stating the differences between these knowledge forms but proceed – very much in the spirit of Bruno Latour – to study the entanglement of these forms of knowing within the context of several volcanic crises. The knowledge forms may be different, even radically different, but during the time of crisis, when scholars seek scientifically justified responses, these forms of knowing get entangled in exciting and productive ways. I think this study and its setting has much to offer to narrative scholars and students of other kinds of crises.

Response: Thanks very much to Professor Hyvarinen for their careful and enthusiastic reading of what we are trying to do here. It is really pleasing to us that we have managed to make our aims clear but also that it is found to be interesting and useful!

The performed narrative analysis is as such impeccable enough but might profit from further subtlety. Words like "expectation", "surprise", and "uncertainty" recur in the text. Jerome Bruner (1990; 1991; cf. Hyvärinen 2016) has theorized narrative as a language and knowledge form whose function is to react to unexpected, surprising, and non-canonical events. He describes how cultures encode the expectations of the normal, proper, and canonical in the form of sequential scripts. The description of how the scientists were observing the gradually exploding volcanos, being insecure about whether the process follows the scientifically processed, expected course of events, or does it and how and when does it violate the expectations, reminds me closely of Bruner's discussion. To add one more layer, many recent narratologists (Fludernik 1996; Herman 2009; Caracciolo 2014) have defined narratives not merely as sequences of events but have accentuated the aspect of experiencing and emotions in the recounted storyworlds. As we read the article and the stories the scientists tell, we can see that intense feelings and disrupting experiences are strong drivers of telling.

Response: this is a very interesting point here. Our principle aim in this moment of 'entanglement' between paradigmatic and narrative forms – is the role that it plays in these moments of uncertainty, and is an important means by which uncertainty can be expressed or reproduced for the listener. We felt that this accentuation of experience and emotions is important and so have included further acknowledgement of this point in our proposed amendment to the introduction.

One laudable move in the article (in 135) was when the interviewers "asked them how, where, when and why they more normally would tell these stories". This is a clever move since interviews have often been criticized for their "artificiality" (in most cases unfairly, but it is another story entirely). Now the researchers wanted to locate the "natural environments" of these stories by putting the interviewees to think about their telling habits. Of course, it is still different from the recording of the storytelling in situ…but when indeed can we be there in right time and place, with a recorder and proper permits? In any case, this move indicates that the researchers have not only collected stories as descriptions of something (say, volcanic crises) but wanted to address the "narrative realities" where the scientists already were telling these influential stories to their colleagues and students (see Gubrium & Holstein 2009).

Also, speaking about the methodology, the idea of asking three different stories sounds elegant and justified, precisely for letting the tellers to express a variety of experiences and interpretations. Of course, the employed thematic reading is not the most elegant or productive way of reading interview materials. In this study, it gives a rough and informative outline of the contents. By paying more attention to the interview interaction and the nuances of language use, this exciting material might offer chances for more thorough and comprehensive analysis. As regards the tools of more nuanced readings, I would recommend several chapters in De Fina & Georgakopoulou 2015 (esp. Parts III-IV). I hope the team will continue its cooperation with this intriguing theme!

Response: we agree, and thank you for the suggested reading here, which was very helpful and inspiring! The 'thematic' approach we employ here has allowed us to more closely match the established scientific and scientific modes of analysis of the drivers as risk as well as combine this with some more basic elements of narrative analysis. This was used to help us achieve the main aims of our paper which was to explore the inherent value of 'storytelling' by scientists in creating new understandings of risk in uncertain situations and how the process of storytelling might help them to cope.

Nonetheless, the references provided here not only stimulate thoughts for further work but also provide some helpful insights into deepening the analysis presented here, particularly in considering the interactivity of the storytelling group we convened, and the 'genres' ( sensu Hyvarinen, 2015) deployed by storytellers both in the content of their interviews but in their imagining of how and why they more usually tell it.

We have amended the second last paragraph of the introduction to include

*Narratives are functionally rooted in experience and the violation of expectations that drive uncertain situations (Hyvärinen, 2016), making them particularly well suited to describing and analysing crisis events. Further, decision-making or actions under conditions of uncertainty, can create discomfort for scientists when responding to these crises and these emotions are not only strong drivers for storytelling but provide outlets for the emotions associated with these experiences. Consequently, in addition to the knowledge in the content of the narrative, the action of storytelling, and its affect are important to understand (Goodwin, 2015).*

And a sentence in the methods section to acknowledge our choice of thematic analysis *(Italicised here)*

Our methodological framework was shaped by our desire to understand how the stories that scientists tell during a crisis can help to shape ideas and knowledge around them: *the thematic analysis was thus particularly shaped by theoretical understandings of the drivers of risk in this context*

This allowed us to set up and explore how the themes described in this review, and signpost where further analysis with these broader tools and ideas would yield even further insight. We propose to embedded this reflection into Section 3.4

*Our thematic analysis of the content of the stories themselves provides some insights into this value, which could be developed even further by drawing more deeply on narrative analysis within these themese (e.g. DeFina and GeorgakoPoulou 2017). Narrative analysis demonstrates that the value of the narrative is shaped by more than the content alone; the interactions between 'storytellers' and their audiences, and the physical and unspoken vocabularies create and shape further knowledge as it is used (Goodwin, 2015). As an illustration of this potential, we noted from our own experience, that during our informal group conversation the storytelling process most closely resembled natural conversations between scientists, it is noteworthy here that the stories here escalated into nine further spontaneous discussion points of direct relevance to responding to volcanic risk (Table 1), involving the sharing and shaping of each other's ideas. However, the fact that most of our storytellers identified the value of swopping and sharing and stories with one another primarily as a means of entertainment or social bonding shows that their inherent value is currently underestimated within our community. New and further work could develop this important avenue by using narrative analysis to improve our framework.*

We also propose adding an additional sentence to our conclusions.

*This methodology has the potential to yield even further insights if deepened to include further dimensions of narrative analysis to include the collection of details of the interactions during the telling of these stories,  as well as more nuanced influences of the languages of storytelling including the genres or modes of storytelling and their impact on content relevant to understanding uncertain crisis situations.*

We will also add the relevant additional references to the list.

Some details:

(40) The classical sociolinguists are William Labov (not Lebov) and Joshua Waletzky (not Waletsky). The philosopher, in (30) is Bruno Latour, not LaTour. (125) Jackson, 2002, is missing from the references.

These typos and omissions have been dealt with.

eferences:

Bruner, J. (1986). *Actual Minds, Possible Worlds*. Cambridge, Ma.: Harvard University Press.

Bruner, J. (1990). *Acts of Meaning*. Cambridge: Harvard University Press.

Bruner, J. (1991). The Narrative Construction of Reality. *Critical Inquiry 18*, 1-21.

Caracciolo, M. (2014). *The Experientiality of Narrative. An enactivist approach*. Berlin: De Gruyter.

De Fina, A., & Georgakopoulou, A. (Eds.). (2015). *The Handbook of Narrative Analysis*. Chichester: Wiley Blackwell.

Fludernik, M. (1996). *Towards a 'Natural' Narratology*. London and New York: Routledge.

Gubrium, J. F., & Holstein, J. A. (2009). *Analyzing Narrative Reality*. Los Angeles, CA, London: Sage.

Herman, D. (2009). *Basic Elements of Narrative*. Malden, MA: Wiley-Blackwell.

Hyvärinen, M. (2016). Expectations and experientiality: Jerome Bruner's "canonicity and breach". *Storyworlds, 9*(1), 1--25.

REVIEWER TWO

This study draws attention to an important component of being a scientist - the inherent desire many scientists have to communicate their knowledge via an organic storytelling mechanism. This examination is an interesting complement to the growing literature on science communication more broadly.

As a scientist largely concerned with the status of the natural environment, I must confess the written style of this particular study is quite distinct from that of the conservation science discipline I am accustomed to reading. As such, I found some of the terminology unfamiliar and at times difficult to distil. I have understood the intent of this study correctly, (the value of storytelling for the teller as much as the listener or receiver) I wonder whether the language used to describe this work will be easily digested by others in the environmental scientific community? Given analytical environmental scientists (i.e. vulcanologists) are the subject of interest, I would caution the use of terminology that is fairly distinct to the social science community. I think the concept underpinning this study is compelling and an important message lies within for all environmental scientists. We should be telling stories about our respective scientific observations and experiences with a range of audiences. The benefits, as illustrated here, are two-fold. My request is to make the story the authors of this study are telling more accessible for their own readers.

Thanks for sharing your review and this important perspective. To address this in the corrected document we have gone carefully through the whole manuscript but have particularly adapted the accessibility of the manuscript to make our core goal of understanding how storytelling frameworks can tell us more about environmental crises. To follow on with your point below we have now also more made the link to other types of crises clearer too – providing a specific example (climate crisis) and more clearing spelling out the value to understanding risk and its social constructs as well as natural events.

In particular we want to share the amended text '). Using the personal and the particular enhances salience and offers the opportunity to draw in listeners who may otherwise tire of the message or messenger (Storr, 2019). For example, narratives are strongly encouraged as means to engage non-scientists in critical messages around a changing climate (Corner et al., 2018, Shepherd et al., 2018).

Thus, part of the focus of this paper is to understand how storytelling functions as a means to both share and to shape knowledge (Jasanoff, 2004, Swedlow 2012, Jackson, 2002). However, our aim here is not just to examine storytelling as a technique for increased comprehension and engagement (Dahlstrom, 2014) but to understand how it serves all participants, in this instance those who tell the stories themselves: the scientists.

 This is because understanding and making sense of the intersection between the natural environment and human cultures generates profound challenges for those also engaged in minimising impacts.  Complexities and heterogeneities across multiple hazard and social systems can obscure pathways to the most effective collection or use of scientific information. The way in which scientists navigate and then make sense of these situations is under-explored, particularly for the role it plays in how they make sense of the crisis, both in the moment and in governing their future actions and understanding. In most situations of environmental risk it is acknowledged that hazard scientists have some power (in that they both implicitly and explicitly hold a seat at the decision-making table during a crisis) but with that power comes an obligation to anticipate what might happen next or to outline 'what to do'. So their lived experience of the crisis may not feel powerful to them or reflect external perceptions of tensions that exist. Official articulations of hazard and risk, such as peer-reviewed papers, only cover part of the insights into crisis response, conforming to expectations around disciplinary boundaries or norms. However, we argue it is in the space where natural and human systems collide that valuable lessons for coping with rapid environmental change lie; and that the analysis of sense-making through storytelling has much to offer, partly because of the fluidity of interpretation implicit in their structure, which matches the uncertain and evolving understandings. This is complementary to the conceptualisation of 'storylines' emerging in the climate science community (Shepherd et al., 2018) where narrative use is highlighted as a means to explore plausible future climates,  or use past events as demonstrators of future scenarios.

We develop this further, and use the specific example of a volcanic crisis, and scientists' stories of the events that surrounded a series of eruptions during that time to explore the value of narratives to sense-making through the lens of an environmental crisis, where decision-making is often dependent on rapidly changing and ambiguous scientific information, on short time-scales. We wanted to understand whether stories told of such moments generate or highlight knowledge and information that do not necessarily appear in more conventional scientific literatures or learnings from an environmental crisis, and how that 'tacit' knowledge shapes future actions and views. . We wanted to understand the various ways that the recounting and recalling of events during volcanic eruptions happen and how this shaped the understanding of the storytellers. In this context we are dealing with a risk 'system' analogous to many other  human-natural risk systems particularly those where the timescale for action is less than the timescale over which the risk and its implications can be fully understood.

Another aspect that may warrant further acknowledgement is how the lessons learned from this work do or do not translate into other environmental crisis scenarios. The authors do speculate this transferability to other scenarios in section 3.5, but I feel this case could be

made earlier in the manuscript to highlight the applied relevance and importance of this study. For example, it could be acknowledged that the story-telling commentary around different environmental crises may vary based on whether the crisis is ultimately anthropogenically derived (i.e. climate change, biodiversity loss) or truly stochastic or natural (i.e. volcanic eruptions). The individual interpretation of 'meaning' or 'emotional effect' from environmental crises may well vary based on the role of blame, accountability or responsibility. This could also differ between individual value systems and the willingness of individuals to take responsibility for their own role in environmental crises.

The reviewer makes some really interesting points here and we have partly embedded these thoughts into the part of the new introduction above. The distinction between anthropogenically stimulated crises and those where social responses to wholly 'natural' hazards is really interesting – particularly in the context of the emotional responses and influence on the scientists involved. This is a very large topic and would require long and further analysis, particularly as our example is so firmly rooted in one area.  So we have not explicitly addressed this. The point is also germane to the other reviewers points about developing a deeper narrative analysis to delve even further, particularly  unspoken narratives and the genre or mode of telling, so we think it is embedded into our new discussion of further analysis, this is particularly important here. This is a new way of thinking about narrative relative to some other existing studies, so signposting ways to widen further work is useful.

I look forward to seeing this study finalised and contributing to the important body of work on the role of environmental scientists as communicators (or storytellers) on topics of general importance for the global community.

We hope that the changes made to the introduction this is now clearer to wider variety of readers, thank you for thoughts on this which undoubtedly helped.

**Citation**: https://doi.org/10.5194/egusphere-2023-71-RC2

Relevant Additional References now in manuscript:

Corner, A. Shaw, C. and Clarke, J. (2018) Principles for effective communication and public engagement on climate change: A Handbook for IPCC authors. Oxford: Climate Outreach.

DeFina, A. and Georgakopoulou, A. (2015). The Handbook of Narrative Analysis. First Edition. Wiley. 454pp.

Goodwin, C., (2015) Narrative as Talk-in – interaction. In: The Handbook of Narrative Analysis, Editors A. DeFina and A. Georgakopoulou. Wiley. 454pp

Hyvarinen, M. (2016) Expectations and experientiality: Jerome Bruner's 'cannonicity and breach'. Storyworlds 9: 1-25.

Monteil, C., Barclay, J. and Hicks, A. (2020) Remembering forgetting, and absencing disasters in the post-disaster recovery process. International Journal of Disaster Risk Science, 11: 287-299

Walshe, R.A., Adamson, G.C.D. and Kelman, I. (2020) Helices of disaster memory: how forgetting and remembering influence tropical cyclone response in Mauritius. International Journal of Disaster Risk Reduction, 50: 101901

Shepherd, T.G., Boyd, E., Calel, R.A., Chapman, S.C., Dessai, S., I.M. Dima-West. Fowler, H.J., James, R., Maraun, D., Martius, O., Senior, C.A., Sobel, A.H., Stainforth, D.A., Tett, S.F.B., Trenberth, K.E., van den Hurk, B.J.J.M., Watkins,N.W., Wilby, R.L., Zenghelis, D.A. (2018) Storylines: an alternative approach to representing uncertainty in physical aspects of climate change. Climatic Change 151: 551-571